# Evaluation of the Effectiveness of *Staphylococcus* Phages in a Skincare Serum against *Staphylococcus* spp.

Wattana Pelyuntha [1], Mingkwan Yingkajorn [2], Thamonwan Narkpao [1], Supanida Saeaui [1], Khemapsorn Promkuljan [1] and Kitiya Vongkamjan [1,*]

1. Department of Biotechnology, Faculty of Agro-Industry, Kasetsart University, Chatuchak, Bangkok 10900, Thailand; wpelyuntha@gmail.com (W.P.); thamonwan.narkpao@gmail.com (T.N.); supanida.saea@gmail.com (S.S.); bbbkhemprom@gmail.com (K.P.)
2. Department of Pathology, Faculty of Medicine, Prince of Songkla University, Hat Yai 90110, Thailand; mingkwan.y@psu.ac.th
* Correspondence: kitiyavongkamjan.a@ku.th

**Abstract:** The emergence of multidrug-resistant (MDR) *Staphylococcus* spp. has resulted in the reduced use of antibiotics in many skincare cosmetic products. Alternative treatments using natural bioactive compounds and chemical agents can be replaced. However, these compounds have induced negative side effects among users and are not environmentally friendly. Phage therapy is an alternative to antibiotics for the treatment of specific pathogenic bacteria including *Staphylococcus* spp., without harmful effects on human skin cells and microflora. Phages can be potentially used in cosmetic products. The direct application of phage-based cosmetic products on skin can reduce the chance of skin infection caused by pathogenic *Staphylococcus* spp. In the present work, we isolated 17 *Staphylococcus* phages from sewage and soil samples. Phage A1 showed the highest lytic ability at 50% (B1 profile), covering 13 tested *Staphylococcus* isolates including *Staphylococcus aureus* (SA), methicillin-resistant *S. aureus* (MRSA), *S. capitis* (SC), and *S. epidermidis* (SE). Phage A1 reduced the representative *S. aureus* ATCC 25923 and *S. capitis* SC1 by $2.0 \pm 0.1$ and $4.1 \pm 0.3$ log units at a multiplicity of infection (MOI) of $10^4$ and by $4.2 \pm 0.2$ and $4.4 \pm 0.5$ log units at a MOI of $10^5$ after 6 h of post-phage treatment. The transmission electron microscope revealed that phage A1 was classified in the order *Caudovirales* of the family *Myoviridae* based on its appearance. Phage A1 showed optimal survival in the presence of a 0.125% ($v/v$) solidant DMH suspension after 3 h of post-treatment. Under a phage skincare serum formulation, the titers of phage A1 were reduced by 0.46 and 0.85 log units after storage at 4 and 25 °C, whereas a reduction of 2.96 log units was also observed after storage at 37° for 90 days. This study provides strong evidence for the effectiveness of phage application in cosmetic skincare serum for the treatment of skin diseases caused by MDR and pathogenic *Staphylococcus* spp. The concept of this study could be advantageous for cosmetic and/or cosmeceutical industries searching for new bioactive ingredients for cosmetic/cosmeceutical products.

**Keywords:** biocontrol agent; cosmetic ingredient; *Myoviridae*; phage therapy; *Staphylococcus* spp.

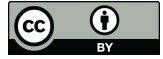

## 1. Introduction

Although *Staphylococcus* spp. live harmlessly on human skin, these Gram-positive cocci bacteria can commonly cause a wide range of clinical infections such as blood poisoning, food poisoning and toxic shock syndrome [1]. Skin infections are also common diseases caused by *Staphylococcus* spp., as this bacterium can enter the wound if the skin is punctured or broken [2]. Infections can present in a variety of ways, such as hair follicle infection (folliculitis), boils, abscesses and sycosis, impetigo, ecthyma, cellulitis, and staphylococcal scalded skin syndrome [3]. Serious skin infections are reported in approximately 14 million U.S. patients annually, with as many as 30% caused by MRSA [4]. To treat *Staphylococcus* infections, multiple antibiotics such as clindamycin, daptomycin, methicillin, vancomycin,

tetracycline, etc., are commonly recommended [5]. However, these bacteria are very adaptable and can become resistant to one or more antibiotics, while serious side effects can also occur during antibiotic administration.

Antimicrobial agents from natural and synthetic sources have been reported to be effective against a wide range of *Staphylococcus* infections, including herbal crude extracts [6–8], essential oils [9–11], organic-acid-based products [12], and antimicrobial peptides [13,14]. However, some essential oils, herbal extracts, and other substances can cause skin irritation if applied in a concentrated form [15]. Importantly, these antimicrobial agents are widely applied in dermocosmetics and/or cosmeceuticals, which has resulted in the arise of non-specific products, especially topical cream/gels and skincare serums. Therefore, other strategies should be discovered to expand the sources of active ingredients for fighting *Staphylococcus* infections, including the use of bacteriophages or bacterial viruses that are well-known as types of "phage therapy".

Bacteriophages (phages) are naturally occurring viruses of bacteria that are present in the environment [16]. Phages can kill specific bacteria without targeting human cells or the surrounding microbiota and are therefore considered safe [17]. Phages have been widely used as an alternative to antibiotics for medical purposes to control target pathogenic bacteria, especially multidrug-resistant (MDR) strains. According to the U.S. FDA, the use of phage therapy has been approved for controlling pathogenic bacteria in foods and the treatment of bacterial infections [18,19]. Phages are an alternative biocontrol due to their specificity, easy administration, and lack of harmful effects on human health [20]. *Staphylococcus* phages have been widely applied for treating methicillin-resistant *S. aureus* (MRSA). For example, three isolated phages, V1SA20, VV1SA19 and V1SA22, are active against 115, 101 and 87 of 150 clinically important *S. aureus* and *S. argenteus* strains that were isolated in France between 2017 and 2020 [21]. A virulent phage, WV, isolated from a slaughterhouse, could lyse MDR-*S. aureus* isolated from clinical patients in China [22]. This phage also destroyed biofilm-producing *S. aureus*. Another study also revealed that the *Staphylococcus* phages vB_SauM-A, vB_SauM-C and vB_SauM-D have an antibiofilm activity. They are more efficient in biofilm removal and reducing the staphylococcal count when compared to antibiotics [23]. In addition, phages can be applied in cosmetic formulations for the efficient treatment of skin diseases caused by *S. aureus* in vitro [24].

The aim of this study was to evaluate the effectiveness of newly isolated phages on *Staphylococcus* spp. strains that cause multiple skin diseases in humans. These phages were isolated from natural sources. The information from this study on the lysis ability of phages will be useful for further phage selection and consideration of the use of phages as an active ingredient in cosmetic products for controlling *Staphylococcus* infections. The concept of this study could be advantageous for cosmetic/cosmeceutical product development in the search for new active ingredients.

## 2. Materials and Methods

### 2.1. Bacterial Strains Used and Culture Condition

Thirteen isolates of *Staphylococcus* were used in this study (Table 1). These included four isolates of *Staphylococcus aureus* (SA1, SA2, SA3 and SA4) that were previously isolated from human skin (acne). Another four isolates of *S. aureus*, SA3503, SA5809, SA1362 and SA2031, previously isolated from the clinical, purulent wound samples of patients and characterized as methicillin-resistant *S. aureus* (MRSA) were kindly provided by the Department of Pathology, Faculty of Medicine, Prince of Songkla University, Songkhla, Thailand. Two isolates of *Staphylococcus capitis* (SC1 and SC2) and *Staphylococcus epidermidis* (PC6 and SE-F1) previously isolated from human skin (acne) were also included. All isolates were propagated in a tryptic soy broth (TSB) at 37 °C for 24 h prior to the studies and kept at −20 °C in 15% (*v/v*) glycerol at the Department of Biotechnology, Faculty of Agro-Industry, Kasetsart University, Chatuchak, Bangkok, Thailand.

**Table 1.** *Staphylococcus* isolates used in this study.

| Bacterial Isolate | Code Name | Source of Origin |
|---|---|---|
| *S. aureus* (*n* = 5) | ATCC25923 | Laboratory collection |
| | SA1 | Human skin (acne) |
| | SA2 | Human skin (acne) |
| | SA3 | Human skin (acne) |
| | SA4 | Human skin (acne) |
| MRSA (*n* = 4) | SA3503 | Purulent wound |
| | SA5809 | Purulent wound |
| | SA1362 | Purulent wound |
| | SA2031 | Purulent wound |
| *S. capitis* (*n* = 2) | SC1 | Human skin (acne) |
| | SC2 | Human skin (acne) |
| *S. epidermidis* (*n* = 2) | PC6 | Human skin (acne) |
| | SE-F1 | Human skin (acne) |

### 2.2. Isolation and Purification of Staphylococcus Phages

Eight wastewater (W) samples and four soil (S) samples were collected from the wastewater treatment station and landfill garden, respectively, at Kasetsart University, Bangkok, Thailand. The wastewater samples were collected using a sterile bottle and kept in an iced box during transportation to the laboratory within 1 h. The soil samples (~0.3 kg) were kept in plastic bags. All samples were stored at 4 °C until processing for phage isolation [25]. Each wastewater sample was centrifuged at 4 °C for 15 min at 6500 rpm. The soil sample (100 g) was mixed with 0.9 L of a salt magnesium (SM) buffer and vigorously shaken in a shaking incubator at 37 °C for 1 h at 220 rpm. The supernatant of each sample was collected and passed through a sterile bottle top filter with a pore size of 0.22 μm (EXTRACTO, GVS Filter Technology, Morecambe, UK). A filtrate was used for phage isolation using a standard double overlay method for the 13 abovementioned isolates. All plates were subjected to incubation at 37 °C for 18 to 24 h. The formation of a district plaque was observed, and it was suspended in the SM buffer for 3-passage purification. A plaque from the third purification passage was used to prepare 10-fold serial dilutions in the SM buffer for another overlay and phage lysate preparation. The phage titer was determined by counting the plaques present on each plate of a given dilution [26].

### 2.3. Host Range Determination

The host range of all the isolated phages was determined by spotting 10 μL of each phage lysate (8 log PFU/mL) on the lawn of a given *Staphylococcus* spp. isolate prepared on the overlay. The plates were observed for the appearance of the clear zone after incubation at 37 °C for 18 to 24 h [26].

### 2.4. Transmission Electron Microscopy of Selected Staphylococcus Phage

A selected phage was analyzed in terms of morphology using a transmission electron microscope (TEM). A copper grid sample was prepared using a given phage lysate at 8 log PFU/mL. Uranyl acetate (1%) was used for negative staining. The image was handled with a TEM model, JEM-2010 (JEOL Ltd., Tokyo, Japan), at 160 kV and an instrumental magnification of 100,000× [25].

### 2.5. Efficiency of Plating of Selected Staphylococcus Phage

A selected phage with the highest % lytic ability was selected for the evaluation of the efficiency of plating (EOP), following the protocol in [25]. Phage A1 was tested three times independently using three dilutions (4–6 log PFU/mL) against 13 *Staphylococcus* spp. The EOP value was calculated using the following formula:

EOP = average PFU on target bacteria/average PFU on host bacteria

The efficiency of plating was classified as "high production" when the ratio was 0.5 or more. An EOP of 0.1 or higher, but below 0.5, was considered as a "medium production" efficiency, and that between 0.001 and 0.1 was considered as a "low production" efficiency. An EOP of 0.001 or below and cases where the dilutions did not result in any plaque formation were classified as "inefficient".

### 2.6. Lytic Ability of Staphylococcus Phage

A selected phage with the highest % lysis based on the host range determination was used for further evaluation. Three isolates of *Staphylococcus* spp. including *S. aureus* ATCC25923, *S. capitis* SC1 and *S. epidermidis* PC6 were used as representative hosts. A suspension of each host presenting an initial cell of 4 log CFU/mL was mixed with a phage at a concentration of 7 and 8 log PFU/mL (MOI $10^4$ and $10^5$, respectively). The culture of each *Staphylococcus* isolate without the phage lysate was kept as a control. The number of *Staphylococcus* cells in the control and phage treatment conditions were evaluated at every 6 h interval for 24 h using a spread plate technique on tryptic soy agar (TSA).

### 2.7. Lytic Activity of Selected Staphylococcus Phage in the Presence of Cosmetic Preservative

The effect of solidant DMH (glydant) on the inactivation of phage activity was investigated after the phage was exposed to the solidant DMH at different concentrations. Briefly, the phage lysate (7 log PFU/mL) was inoculated with 5 mL of solidant DMH at various concentrations (0.1%, 0.125%, 0.25%, 0.5% and 1%) for 3 h. The phage-solidant DMH suspension was centrifuged at 4 °C for 15 min at 12,000 rpm to remove the aqueous phase. The remaining phages were resuspended with 200 μL of SM buffer, followed by the preparation of serial dilutions and counting of the remaining titers on the lawn of a given host strain on a TSA plate in a plaque-forming assay.

### 2.8. Formulation of Skincare Serum Containing Selected Staphylococcus Phage

A cosmetic serum was prepared using sterile DI water (93.675%), hydroxyethyl cellulose (HEC, 1.2%), glycerine (3%) and solidant DMH (0.125%) according to the cosmetic ingredient supplier's formulation (CHEMIPUN Corporation Ltd., Bangkok, Thailand) with slight modifications. Briefly, HEC was combined with DI water (Phase I) and agitated at 60 °C until the base serum was sticky and cooled to room temperature. Glycerol and solidant DMH (Phase II) were mixed until homogeneity was achieved. The phage lysate was combined with the mixture of phases I and II to reach the final concentration of 8 log PFU/mL. The cosmetic skincare serum was transferred to amber glass bottles and stored at 4, 25 and 37 °C for 90 days for the phage survivability test. The viscosity index and % torque of the formulated skincare serum were analyzed using a viscometer (DV2TLVTJ0, Brookfield Viscometer, Middleboro, MA, USA) with an LV-03 probe (12 rpm, 25 °C) using a commercial service provided by the MUPY Dermocosmetic Testing Center, Faculty of Pharmacy Mahidol University, Bangkok, Thailand.

### 2.9. Survivability of Staphylococcus Phage in Skincare Serum

The skincare serum containing the phage stored in the amber glass bottles at 4, 25 and 37 °C was evaluated for phage survivability at days 0, 14, 30, 60, and 90 of storage by preparing 10-fold serial dilutions with SM buffer. The phage titer determination on the lawn of a given host strain (*S. capitis* SC1) on the TSA plate was as described above.

### 2.10. Statistical Analysis

A statistical analysis was performed using SPSS (Version 22.0) of Windows statistics software (SPSS Inc., Chicago, IL, USA). Data regarding the reduction in the *Staphylococcus* count from the lytic ability test and the phage's survivability in the skincare serum were subjected to one-way analysis of variance followed by Tukey's range test. A significant difference between the control and treatment groups was calculated using an independent-

sample *t*-test. A difference was also considered statistically significant at a *p*-value of less than 0.05.

## 3. Results

### 3.1. Isolation of Staphylococcus Phages

A total of 17 *Staphylococcus* phages were isolated from the collected wastewater and soil samples (Table 2). Of these, 15 phages were recovered from eight wastewater samples, whereas two phages (A2 and A8) were recovered from the same soil sample (S1) on two different *Staphylococcus* hosts (SA3 and SA2031). Each wastewater sample yielded between one and three phages on 10 different *Staphylococcus* hosts. Two wastewater samples, W3 and W6, yielded the highest number of phages on three different *Staphylococcus* hosts. Among the four types of *Staphylococcus* hosts, SA and MRSA were the common hosts for phage isolation. The isolated phages showed a plaque size ranging from 0.4 to 1.3 mm in diameter.

**Table 2.** Host of isolation, source of origin and plaque morphotype of isolated phages.

| *Staphylococcus* Phage | Host of Isolation | Source of Origin [1] | Plaque Morphotype |
|---|---|---|---|
| A1 | SC1 | W1 | 1.3 mm |
| A2 | SA3 | S1 | 0.6 mm |
| A3 | ATCC25923 | W2 | 0.6 mm |
| A4 | SA2 | W3 | 0.5 mm |
| A5 | SA5809 | W5 | 0.4 mm |
| A6 | SA5809 | W6 | 0.4 mm |
| A7 | SA1362 | W3 | 0.5 mm |
| A8 | SA2031 | S1 | 0.6 mm |
| A9 | SA1 | W6 | 0.5 mm |
| A10 | SC1 | W6 | 0.4 mm |
| A11 | SA3 | W7 | 0.5 mm |
| A12 | SC2 | W4 | 0.5 mm |
| A13 | SC2 | W5 | 0.4 mm |
| A14 | SA4 | W7 | 0.4 mm |
| A15 | SA4 | W1 | 0.4 mm |
| A16 | SA4 | W8 | 0.4 mm |
| A17 | SA4 | W3 | 0.6 mm |

[1] W: wastewater sample; S: soil sample.

### 3.2. Host Range Determination of Staphylococcus Phages

The isolated phages showed 12 different lytic profiles on 13 isolates of *Staphylococcus* (Table 3). Phage A1 showed the highest lytic ability (profile B1) against five *S. aureus* and two *S. capitis* isolates (53.8% lysis), followed by phage A2 (profile N1; 23.1% lysis). In addition, the phages only showed specific lysis against their natural host of isolation, presenting a 7.7% lytic ability (profiles N2–N11). Although several phages showed identical lysis profiles, these phages were isolated from different samples. For example, profile N11 was observed in four phages isolated from four different samples, but these shared the same host of isolation.

**Table 3.** Lysis profiles of *Staphylococcus* phages.

| Tested Isolates | *Staphylococcus* Phages [1] | | | | | | | | | | | | | | | | |
|---|---|---|---|---|---|---|---|---|---|---|---|---|---|---|---|---|---|
| | A1 | A2 | A3 | A4 | A5 | A6 | A7 | A8 | A9 | A10 | A11 | A12 | A13 | A14 | A15 | A16 | A17 |
| ATCC25923 | + | − | + | − | − | − | − | − | − | − | − | − | − | − | − | − | − |
| SA1 | + | + | − | − | − | − | − | − | + | − | − | − | − | − | − | − | − |
| SA2 | + | − | − | + | − | − | − | − | − | − | − | − | − | − | − | − | − |
| SA3 | + | + | − | − | − | − | − | − | − | − | + | − | − | − | − | − | − |
| SA4 | − | − | − | − | − | − | − | − | − | − | − | − | − | + | + | + | + |
| SA3503 | − | − | − | − | − | − | − | − | − | − | − | − | − | − | − | − | − |
| SA5809 | − | − | − | − | − | + | + | − | − | − | − | − | − | − | − | − | − |
| SA1362 | − | − | − | − | − | − | + | − | − | − | − | − | − | − | − | − | − |
| SA2031 | + | − | − | − | − | − | − | + | − | − | − | − | − | − | − | − | − |
| SC1 | + | − | − | − | − | − | − | − | − | + | − | − | − | − | − | − | − |
| SC2 | + | + | − | − | − | − | − | − | − | − | − | + | + | − | − | − | − |
| PC6 | − | − | − | − | − | − | − | − | − | − | − | − | − | − | − | − | − |
| SE-F1 | − | − | − | − | − | − | − | − | − | − | − | − | − | − | − | − | − |
| %Total lysis | 53.8 | 23.1 | 7.7 | 7.7 | 7.7 | 7.7 | 7.7 | 7.7 | 7.7 | 7.7 | 7.7 | 7.7 | 7.7 | 7.7 | 7.7 | 7.7 | 7.7 |
| Profile Code | B1 | N1 | N2 | N3 | N4 | N4 | N5 | N6 | N7 | N8 | N9 | N10 | N10 | N11 | N11 | N11 | N11 |

[1] Green area indicates lysis and gray area indicates non-lysis.

### 3.3. Transmission Electron Microscopy of Selected Staphylococcus Phage

An image from the TEM analysis indicated that phage A1 had an icosahedral head, connected by an apparent collar to a helical contractile tail (Figure 1). These morphological features suggested that A1 belongs to the family *Myoviridae* of the order *Caudovirales*. Based on these classifications, phage A1 was renamed as vB_SenM_A1, following the International Committee on the Taxonomy of Viruses (ICTV) guidelines.

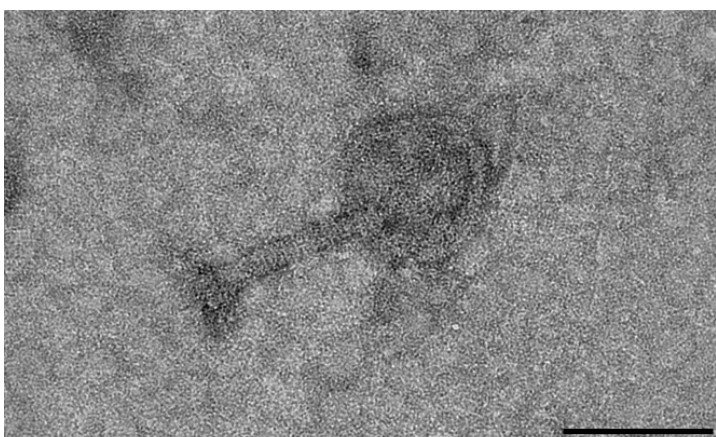

**Figure 1.** Morphology of *Staphylococcus* phage A1 (100,000× magnifications) under TEM analysis. Bars represent the length of 100 nm.

### 3.4. Efficiency of Plating of Selected Staphylococcus Phage

The EOP assay revealed that phage A1 showed high production on four staphylococcal isolates, medium production on three isolates, and inefficient production on six isolates. The EOP values are shown in Table 4.

**Table 4.** The efficiency of plating (EOP) of phage A1.

| *Staphylococcus* spp. | EOP Value [1] | Category [2] |
|---|---|---|
| ATCC25923 | 0.81 ± 0.07 | High |
| SA1 | 0.23 ± 0.09 | Medium |
| SA2 | 0.71 ± 0.16 | High |
| SA3 | 0.44 ± 0.05 | Medium |
| SA4 | <0.001 | Inefficient |
| SA3503 | <0.001 | Inefficient |
| SA5809 | <0.001 | Inefficient |
| SA1362 | <0.001 | Inefficient |
| SA2031 | 0.47 ± 0.02 | Medium |
| SC1 | 1.00 * | High |
| SC2 | 1.60 ± 0.21 | High |
| PC6 | <0.001 | Inefficient |
| SE-F1 | <0.001 | Inefficient |

[1] All values are provided as the mean ± standard deviation of triplicates ($n = 3$). [2] EOP value: high production (EOP > 0.5), medium production (0.1 < EOP < 0.5), low production (0.001 < EOP < 0.1) and inefficient production (EOP < 0.001). * The original strain of isolation has an EOP value of 1.00 and is shown in boldface (*S. capitis* SC1 for phage A1).

### 3.5. Lytic Ability of Staphylococcus Phage A1

The *Staphylococcus* phage A1, at MOI $10^4$, could significantly ($p < 0.05$) reduce the cell numbers of *S. aureus* ATCC25923 and *S. capitis* SC1 by 2 log units and 4.1 log units, respectively, after 6 h of phage treatment initiation (Table 5). After 24 h, the cell counts of *S. aureus* ATCC25923 and *S. capitis* SC1 in the control reached as high as 8.5 and 8.7 log CFU/mL, respectively. However, treatments with phage A1 led to the remaining cell count of 3.4 log CFU/mL for *S. aureus* ATCC25923 and no cells for *S. capitis* SC1. At a higher MOI ($10^5$), the cell counts of both isolates were lower than the detection limit (completely eliminated) after 6 h of phage treatment. After 6 h, phage A1, at a higher MOI, showed 100% reduction or presented a cell reduction > 4 log units when compared to the control. In contrast, phage A1, at both MOIs, could not decrease the number of *S. epidermidis* PC6, as the cell count reached a similar level to the control (over 8 log CFU/mL) after 24 h of the study.

**Table 5.** Efficacy of *Staphylococcus* phage A1 on representative *Staphylococcus* spp.

| Strains | Time (h) | Bacterial Count (log CFU/mL) [1] | | |
|---|---|---|---|---|
| | | Control | MOI $10^4$ | MOI $10^5$ |
| *S. aureus* ATCC25923 | 0 | 4.1 ± 0.7 [a] | 4.3 ± 0.3 [a] | 4.2 ± 0.2 [a] |
| | 6 | 6.7 ± 0.8 [b] | 2.3 ± 0.2 [c*] | 0.0 ± 0.0 [b*] |
| | 12 | 7.2 ± 0.7 [c] | 1.0 ± 0.3 [d*] | 0.0 ± 0.0 [b*] |
| | 18 | 8.2 ± 0.4 [d] | 2.4 ± 0.6 [c*] | 0.0 ± 0.0 [b*] |
| | 24 | 8.5 ± 0.3 [d] | 3.4 ± 0.2 [b*] | 0.0 ± 0.0 [b*] |
| *S. capitis* SC1 | 0 | 4.3 ± 0.5 [a] | 4.1 ± 0.3 [a] | 4.4 ± 0.5 [a] |
| | 6 | 7.3 ± 0.2 [b] | 0.0 ± 0.0 [b*] | 0.0 ± 0.0 [b*] |
| | 12 | 7.9 ± 0.4 [bc] | 0.0 ± 0.0 [b*] | 0.0 ± 0.0 [b*] |
| | 18 | 8.2 ± 0.3 [c] | 0.0 ± 0.0 [b*] | 0.0 ± 0.0 [b*] |
| | 24 | 8.7 ± 0.3 [d] | 0.0 ± 0.0 [b*] | 0.0 ± 0.0 [b*] |
| *S. epidermidis* PC6 | 0 | 4.3 ± 0.2 [a] | 4.2 ± 0.1 [a] | 4.0 ± 0.5 [a] |
| | 6 | 6.4 ± 0.2 [b] | 6.1 ± 0.3 [b] | 6.2 ± 0.2 [b] |
| | 12 | 7.3 ± 0.4 [c] | 7.4 ± 0.1 [c] | 7.0 ± 0.4 [c] |
| | 18 | 7.9 ± 01 [d] | 8.0 ± 0.3 [c] | 7.8 ± 0.3 [d] |
| | 24 | 8.5 ± 0.3 [e] | 8.7 ± 0.6 [d] | 8.2 ± 0.2 [d] |

[1] All values are provided as the mean ± standard deviation of triplicates ($n = 3$). Different lowercase letters indicate a significant difference ($p < 0.05$) in the cell counts for a given treatment or control between each sampling time. The asterisk (*) indicates a significant difference ($p < 0.05$) between the control and phage treatments at the same time.

### 3.6. Survivability of Staphylococcus Phage in the Presence of Cosmetic Preservative

Phage A1, in the presence of low concentrations of the solidant DHM at 0.10 and 0.125% (*v/v*), could survive at the level of 100% (>7.0 ± 0.1 log PFU/mL) and 99.72 ± 0.23% (7.0 ± 0.1 log PFU/mL), respectively. However, concentrations of the solidant DHM of 0.25 to 1% (*v/v*) could completely kill the phage, as shown in Table 6.

**Table 6.** Survivability of *Staphylococcus* phage A1 in the presence of solidant DHM.

| Solidant DHM Concentration (% *v/v*) | Phage Titers (log PFU/mL) [1] | % Survivability [1] |
|---|---|---|
| 0.0 (DI control) | 7.0 ± 0.2 | - |
| 0.10 | 7.0 ± 0.1 | 100.0 ± 0.0 |
| 0.125 | 7.0 ± 0.1 | 99.7 ± 0.2 |
| 0.25 | 0.0 ± 0.0 | 0.0 ± 0.0 |
| 0.50 | 0.0 ± 0.0 | 0.0 ± 0.0 |
| 1.00 | 0.0 ± 0.0 | 0.0 ± 0.0 |

[1] All values provided as the mean ± standard deviation of triplicates (*n* = 3).

### 3.7. Formulation of Skincare Serum Containing Staphylococcus Phage

Under visual observation, the formulated skincare serum containing *Staphylococcus* phage presented a semi-solid texture and was colorless, as shown in Figure 2. The viscosity index and % torque of the formulated serum were 9164.3 ± 104.5 cps and 91.6 ± 1.0, respectively.

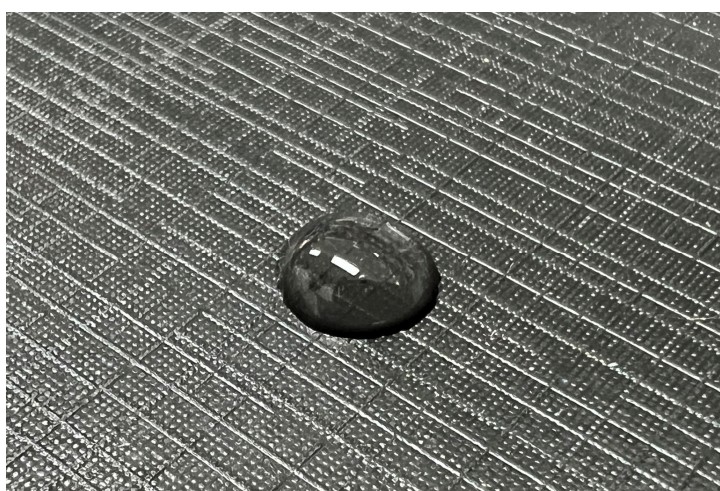

**Figure 2.** Appearance of the skincare serum containing *Staphylococcus* phage A1.

### 3.8. Survivability of Staphylococcus Phage in Skincare Serum

Phage A1 showed slight decreases in the phage titers of 0.46 and 0.85 log units (from 8.08 ± 0.21 to 7.62 ± 0.46 and 8.09 ± 0.17 to 7.24 ± 0.17 log PFU/g) during 90 days of storage at 4 and 25 °C, respectively (Figure 3). A significant difference between the phage titers was observed after storage at 4 or 25 °C for at least 60 days. Similarly, the significant reduction in the amount of phage A1 in the skincare serum started to occur when it was stored at 37 °C for 60 days, while a reduction of up to 3 log units (from 8.07 ± 0.25 to 5.11 ± 0.10 log PFU/g) was observed after storage at 37 °C for 90 days ($p < 0.05$).

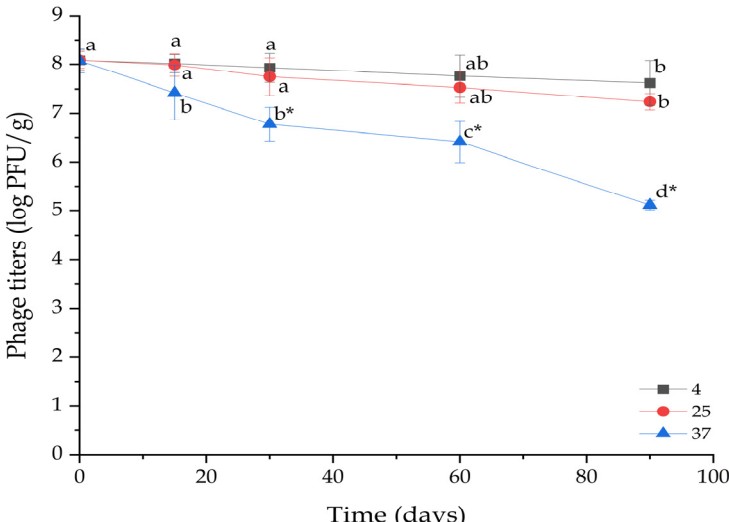

**Figure 3.** Survivability of *Staphylococcus* phage A1 in skincare serum. Different lowercase letters indicate a significant difference ($p < 0.05$) in the phage titers between storage days at a given temperature. The asterisk (*) indicates a significant difference ($p < 0.05$) in the phage titers at different temperatures on a given storage day.

## 4. Discussion

### 4.1. Sources and Lytic Ability of Staphylococcus Phages

*Staphylococcus* phages can be recovered from diverse sources including sewage [27], wastewater [28–30] and human and animal sources [31]. Several studies have reported that *Staphylococcus* phages can be isolated from different types of water samples and used as bactericidal agents against SA, MRSA [27,29,31], *S. haemolyticus*, *S. saprophyticus*, SE [30] and 29 multi-species of *Staphylococci* [28]. For example, *Staphylococcus* phage Spa-3 was previously isolated from a sewage sample in South Korea and later used for targeting skin lesions caused by clinically important *S. aureus* [27]. Phage SaGU1 was also isolated from sewage in Gifu, Japan. This phage effectively killed clinical *S. aureus* isolated from patients with atopic dermatitis but did not kill *S. epidermidis*, a symbiotic bacterium in found skin microflora [32]. Four virulent phages in a developed phage cocktail, namely, "APTC-C-SA01", were recovered from the nasal swabs of patients with chronic rhinosinusitis (APTC-SA-2, APTC-SA4), a soil sample (APTC-SA-12) and animal feces (APTC-SA-13) [31]. In the current study, with a small number of wastewater and soil samples, up to 17 phages were recovered from the same sources as those previously reported. Most studies reported that SA and MRSA are common host species for phage isolation, and we also observed that most of our phages were isolated on SA and MRSA. This must be due to the high prevalence of SA and MRSA in environmental samples, especially in wastewater and sewage [33,34]. This evidence confirmed that most *Staphylococcus* phages normally exist where their bacterial hosts exist [35,36], while SE is a commensal bacterium ubiquitously present on human skin [37].

In our study, only phage A1 was shown to have a broad host range, exhibiting lysis mostly on SA and SC isolates. In addition, this phage did not show any lysis on three of the four tested MRSA. Conventionally, most isolated phages are reported as species-specific, with a narrow host range for a specific *Staphylococcus* isolate. In a previous study, Göller et al. showed that their 90 *Staphylococcus* phages isolated from various wastewater samples in treatment plants located in Zürich, Switzerland, showed different broad host ranges of lysis on a diverse panel of 117 *Staphylococcus*, representing 29 species, including SA, SE, SC, *S. hyicus*, *S. intermedius*, *S. saprophyticus*, *S. sciuri*, etc. [28]. In the same study, only four phages extensively infected a single species. Moreover, seventy-eight *Staphylococcus* phages from sewage effluents were capable of lysis against an international collection of 185 isolates of *S. aureus*, representing MRSA and methicillin-susceptible *S. aureus* (MSSA)

clonal complexes [29]. In our study, we did not have any phages targeting SE. This might be due to the narrow host lysis ability of our isolated phages. Our study is consistent with Shimamori et al. (2021), who found that the phage SuGU1 did not affect *S. epidermidis*, one of the most typical beneficial bacteria [32]. However, novel lytic phages explored in another study can eliminate *S. epidermidis* causing chronic prosthetic joint infections associated with biofilm growth [38].

Generally, phages exhibiting the highest % lytic ability on their target hosts are preferred for use as biocontrol agents for several purposes [39–41]. In the current study, only phage A1 was chosen as a representative for further development based on its lytic capability. Further exploration will still be needed to obtain additional phages with a broader host range. However, for the potential development of a phage cocktail against a wide range of species of *Staphylococcus*, individual phages that have shown lytic activity on other specific species may be included.

Phage A1 was shown to be effective against three representative species of *Staphylococcus*, especially at a higher MOI. The complete elimination of *Staphylococcus* populations was observed. In a previous study, *Staphylococcus* phage JD419 could decrease a number of *Staphylococcus* by 2–3 log units at MOI of $10^1$ and $10^2$ [42]. At the same MOI of $10^2$, the phages phiIPLA-RODI and phiIPLA-C1C, applied against planktonic staphylococcal cells, instigated a reduction in the number of staphylococcal cells of 5 log units at 8 h post-phage treatment [43]. A higher MOI explored in another study of phage SaGU1 showed a reduction of up to 4 log after 9–13 h [32]. The use of phages in the reduction in the level of bacteria depends on the initial phage concentration on the target bacterial cells. A high dose/concentration of phages is strongly recommended if the bacteria have a short generation time or are fast-growing. Overall, in our study, we demonstrated that high MOIs between $10^4$ and $10^5$ have a superior capacity to infect bacterial cells simultaneously.

*4.2. Cosmetic Formulation Containing Potential Phages Targeting Skin Diseases Caused by Staphylococcus spp.*

Phages have recently been introduced as skincare therapeutics and, importantly, as an alternative to antibiotic-based products. Acne vulgaris is the most skin disease, which manifests as *Cutibacterium acnes* (*C. acnes*) [44,45]. It is the most common target for phage-based topical product development. Phages can also induce immunity by helping or improving the immune responses of mammal cells to eliminate bacterial pathogens [46–49]. Therefore, the potential application of phages in acne treatment could change the concept of acne, as inflammation appears to play a prominent role in its pathology [47,50]. In our study, phage A1 was not tested for lytic activity on *C. acnes*. We believe that each natural phage has its own particular host range [51]. Phage A1 is very species-specific with regard to its host and only infects a single species or specific strains within a species, as previously mentioned. Hence, phage A1 might not be involved in the inhibition of acne vulgaris development caused by *C. acnes*.

*Staphylococcus* phages have previously been developed in cosmetic formulations. Phages IS-1 to IS-4 can reduce bacterial growth by 95.45%, compared with free phages and non-supplemented cosmetics, for which the levels are 86.1% and 77%, respectively [24]. In the current study, phage A1 could be combined with the skincare serum, as it showed an ability to reduce the number of SA and SC causing skin diseases in vitro. Phage A1 also survived in the presence of cosmetic ingredients for 90 days of storage. Storage conditions at various temperatures are commonly used as a predictor of the long-term stability of cosmetic products. If a product exhibits acceptable stability and the active ingredient remains at a high temperature, the product will be stable at room temperature for at least two years [52].

We propose that phage A1 is used together with phages against *C. acnes* for synergistic effects, which can be explored in the future. Our results provide valuable data suggesting the potential use of phages in a skincare serum form as a new cosmetic and/or cosmeceutical formulation for controlling pathogenic *Staphylococcus* that cause *Staphylococcus* infection.

## 5. Conclusions

In this study, we obtained 17 *Staphylococcus* phages from environmental sources. A major phage candidate was selected based on its lytic ability against pathogenic *Staphylococcus* species that cause skin diseases. The phages obtained here could be incorporated into cosmetic formulations without affecting their lytic capability due to the ingredients included as cosmetic preservatives. The phages also survived in the cosmetic formulation under the appropriate storage conditions (4–25 °C). This study could be advantageous for the cosmetic and/or cosmeceutical industries, searching for a new bioactive ingredient in cosmetic/cosmeceutical products. Overall, a phage-based cosmetic prototype was the major outcome of this study. However, additional studies need to be established to further explore the stability, shelf-life, safety and efficacy of this product in human trials. These studies will be the focus of our future work.

**Author Contributions:** Conceptualization, W.P. and K.V.; methodology, W.P. and K.V.; investigation, W.P., T.N., S.S. and K.P.; resources, M.Y. and K.V.; data curation, W.P., S.S. and K.P.; writing—original draft preparation, W.P.; writing—review and editing, K.V.; supervision, W.P. and K.V.; project administration, W.P.; funding acquisition, K.V. All authors have read and agreed to the published version of the manuscript.

**Funding:** This research received no external funding.

**Institutional Review Board Statement:** Not applicable.

**Informed Consent Statement:** Not applicable.

**Data Availability Statement:** The data presented in this study are available on request from the corresponding author.

**Acknowledgments:** The authors acknowledge the Department of Biotechnology, Faculty of Agro-Industry, Kasetsart University, Chatuchak, Bangkok, Thailand, and the Department of Pathology, Faculty of Medicine, Prince of Songkla University, Songkhla, Thailand, for the instruments and facility used to conduct this research.

**Conflicts of Interest:** The authors declare no conflict of interest.

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
