# Peer review of "Evaluation of the Effectiveness of Staphylococcus Phages in a Skincare Serum against Staphylococcus spp."

_cosmetics, doi:10.3390/cosmetics10060156_

Round 1

Reviewer 1 Report

Comments and Suggestions for Authors

With respect to the Transmission Electron Microscope of Selected Staphylococcus Phage in results section , please use actual bacteriophage taxonomy

Bacteriophage should be sequenced and seq should be analysed 

the Abstract should have more flow 

some antibiofilm analysis will be beneficial (https://doi.org/10.3390/microorganisms11092352)

in introduction section comparison of bacteriophage activity vs antibiotic activity should be descrubed
https://www.mdpi.com/1422-0067/23/3/1274

discussion section should be slighty rewritten to have mre flow

Comments on the Quality of English Language

Moderate editing of English language required

Author Response

With respect to the Transmission Electron Microscope of Selected Staphylococcus Phage in the results section, please use actual bacteriophage taxonomy

Response: Thank you for your suggestion. Could it be possible for us to keep the current classification based on its tail and head?

Bacteriophage should be sequenced and seq should be analyzed.

Some antibiofilm analysis will be beneficial (https://doi.org/10.3390/microorganisms11092352)

Response: The WGS and antibiofilm studies of phage A1 will be on our pipeline in future works to ensure this phage does not carry lysogeny, toxin, and antibiotic-resistant genes and is safe for cosmetic application. We intend to use a phage cocktail from three or more phages to increase the efficacy of phage application. From this direction, we need more steps of investigation, these results will be published in journals in which our results are suitable for their aims and scopes.

We highly thank the reviewer for the valuable suggestion.

The abstract should have more flow

Response: The Abstract was revised according to the reviewer’s suggestion.

In the introduction section comparison of bacteriophage activity vs antibiotic activity should be described https://www.mdpi.com/1422-0067/23/3/1274

Response: The reference paper that the reviewer provided is very interesting. We included the reference in our text according to the reviewer’s suggestion [Line 72-75].

The discussion section should be slightly rewritten to have more flow

Response: Some part of the Discussion section was revised according to the reviewer’s suggestion.

Reviewer 2 Report

Comments and Suggestions for Authors

In this manuscript, Wattana Pelyuntha and colleagues explored 17 isolated Staphylococcus Phages which are derived from sewage and soil samples, they studied their lytic ability on representative Staphylococcus spp. isolated from clinical samples such as acne and wound, in-vitro efficiency test and TEM analysis showed that Phage A1 had the highest % lytic ability, and Phage A1 was classified to the order Caudovirales of the family Myoviridae based on the appearances. Under the presence of 0.125% (v/v) solidant DMH suspension after 3 h of post-treatment, phage A1 showed optimum survival, importantly, they tested the concentration change of phage A1 in different temperatures under phage-skincare serum formulation. I found this work is very interesting and quite valuable for skincare product development and optimization which for the treatment of skin diseases caused by MDR and pathogenic Staphylococcus spp. Overall, the study design is reasonable, and the experiments were performed excellently. However, there are some points that the authors can optimize to make this manuscript better. I detail my opinions below:

1.      The authors tested the survivability of staphylococcus phage in skincare serum under different temperatures (4, 25 and 37 degrees) and also different time points, here I think the authors should explain the significance of these experiments, which is very necessary because they guided what temperature and how long to store the phage-included skincare serum product is ideal. I also suggest author briefly discuss this part in the Discussion.

2.      Since it was manifested that Cutibacterium acnes(C.acnes) plays a very important role in Acne vulgaris, authors can discuss how phage A1 can directly or indirectly affect Acne vulgaris development.

3.      In Figure 1, the image is not quite clear, it would be better if the authors could provide the zoom-in images that show the icosahedral head and the helical contractile tail.

4.      References: When authors cite published papers, they should describe that reference separately instead of integrating too many (more than 5 paper) references in one sentence, some sentences should be improved, such as the description in lines 57, 284, and 286.

Author Response

In this manuscript, Wattana Pelyuntha and colleagues explored 17 isolated Staphylococcus Phages which are derived from sewage and soil samples, they studied their lytic ability on representative Staphylococcus spp. isolated from clinical samples such as acne and wound, in-vitro efficiency test and TEM analysis showed that Phage A1 had the highest % lytic ability, and Phage A1 was classified to the order Caudovirales of the family Myoviridae based on the appearances. Under the presence of 0.125% (v/v) solidant DMH suspension after 3 h of post-treatment, phage A1 showed optimum survival, importantly, they tested the concentration change of phage A1 in different temperatures under phage-skincare serum formulation. I found this work is very interesting and quite valuable for skincare product development and optimization which for the treatment of skin diseases caused by MDR and pathogenic Staphylococcus spp. Overall, the study design is reasonable, and the experiments were performed excellently. However, there are some points that the authors can optimize to make this manuscript better. I detail my opinions below:

  1. The authors tested the survivability of staphylococcus phage in skincare serum under different temperatures (4, 25 and 37 degrees) and also different time points, here I think the authors should explain the significance of these experiments, which is very necessary because they guided what temperature and how long to store the phage-included skincare serum product is ideal. I also suggest author briefly discuss this part in the Discussion.

Response: The discussion was added according to the reviewer’s suggestion [Line 357-360]

  1. Since it was manifested that Cutibacterium acnes (C. acnes) plays a very important role in Acne vulgaris, authors can discuss how phage A1 can directly or indirectly affect Acne vulgaris development.

Response: The discussion on phage A1 on acne vulgaris caused by C. acnes was added according to the reviewer’s suggestion [Line 346-351].

  1. In Figure 1, the image is not quite clear, it would be better if the authors could provide the zoom-in images that show the icosahedral head and the helical contractile tail.

Response: We apologize for the low quality of Figure 1. We got only two images from the TEM study due to the limitation provided by the commercial service company. The selected image is the highest resolution that we obtained. We apologize for our mistake. However, we have inserted another image for your consideration (PDF file).

  1. References: When authors cite published papers, they should describe that reference separately instead of integrating too many (more than 5 paper) references in one sentence, some sentences should be improved, such as the description in lines 57, 284, and 286.

Response: References were improved according to the reviewer’s suggestion [Line 50-52, 282-283, 285-286].

Round 2

Reviewer 1 Report

Comments and Suggestions for Authors

the authors have made all suggested corrections in the text

Comments on the Quality of English Language

Minor editing of English language required

Reviewer 2 Report

Comments and Suggestions for Authors

My comments have been significantly addressed by authors.